# The genetic legacy of legendary and historical Siberian chieftains

Vincent Zvénigorosky [1,2,12✉], Sylvie Duchesne [3,4,12], Liubomira Romanova[3,12], Patrice Gérard [3], Christiane Petit[5], Michel Petit[5], Anatoly Alexeev[6], Olga Melnichuk [7], Angéla Gonzalez[2], Jean-Luc Fausser[2], Aisen Solovyev[8,9], Georgii Romanov [8,10], Nikolay Barashkov [8,10], Sardana Fedorova [8,10], Bertrand Ludes[1,11], Eric Crubézy[3] & Christine Keyser[1,2]

Seventeen years of archaeological and anthropological expeditions in North-Eastern Siberia (in the Sakha Republic, Yakutia) have permitted the genetic analysis of 150 ancient (15th-19th century) and 510 modern individuals. Almost all males were successfully analysed (Y-STR) and this allowed us to identify paternal lineages and their geographical expansion through time. This genetic data was confronted with mythological, historical and material evidence to establish the sequence of events that built the modern Yakut genetic diversity. We show that the ancient Yakuts recovered from this large collection of graves are not representative of an ancient population. Uncommonly, we were also able to demonstrate that the funerary preference observed here involved three specific male lineages, especially in the 18th century. Moreover, this dominance was likely caused by the Russian conquest of Siberia which allowed some male clans to rise to new levels of power. Finally, we give indications that some mythical and historical figures might have been the actors of those genetic changes. These results help us reconsider the genetic dynamics of colonization in some regions, question the distinction between fact and myth in national histories and provide a rare insight into a funerary ensemble by revealing the biased process of its composition.

[1] CNRS FRE 2029 BABEL, Paris Descartes University, Paris, France. [2] Strasbourg Institute of Legal Medicine, Strasbourg, France. [3] CNRS UMR 5288 AMIS, Toulouse, France. [4] National Institute of Preventive Archaeological Research (INRAP), Cesson-Sévigné, France. [5] French Archaeological Mission in Eastern Siberia (MAFSO), Toulouse, France. [6] Institute of Humanities and Issues of the Minority Peoples of the North, Siberian Branch of the Russian Academy of Sciences, Yakutsk, Russia. [7] M.K. Ammosov North-Eastern Federal University, Yakutsk, Russia. [8] Laboratory of Molecular Biology, Institute of Natural Sciences, M.K. Ammosov North-Eastern Federal University, Yakutsk, Russia. [9] Institute for Humanitarian Studies and Problems of Indigenous Peoples of the North, Yakutsk, Russia. [10] Laboratory of Molecular Genetics, Yakut Science Centre of Complex Medical Problems, Yakutsk, Russia. [11] Paris Institute of Legal Medicine, Paris Descartes University, Paris, France. [12] These authors contributed equally: Vincent Zvénigorosky, Sylvie Duchesne, Liubomira Romanova. ✉email: z.vincent@live.fr

Yakutia is inhabited by half a million Russians, half a million Yakuts (or Sakhas) and a few thousand members of ethnic minorities belonging mostly to the Tungus peoples, that occupied Yakutia centuries before the Yakut period[1]. Traditionally, Yakut culture depends on horse and cattle breeding, whereas the Tungus depend on hunting, gathering and reindeer-herding. In the early seventeenth century, Cossacks serving the Russian Empire reached Siberia and waged war against the Yakuts and Tungus, eventually defeating them and incorporating the region[2]. Christianisation followed the first contacts and neither the Bolshevik revolution nor the Second World War spared Yakutia. These events fundamentally altered certain cultural practices and caused major changes in demographic dynamics and the occupation of the territory.

Archaeological excavations in the Sakha Republic (Yakutia) in North-Eastern Siberia have recovered the remains of 78 men, 51 of which were subjected to genetic analyses in previous studies focused on biogeographic origins[3] or kinship between graves[4–6] using autosomal short tandem repeat (STR) and mitochondrial (HV1) data. Further studies of biogeographic origins also used Y-STR data[7] and finally fragments of the genomes of smallpox[8] and tuberculosis[9] were amplified from some of the same subjects. Anthropological expeditions in Yakutia collected biological samples for more than 200 men, some also subjected to genetic analyses (Y-STR typing) aimed at identifying relationships between the Yakuts and other Siberian populations[10]. These published data (189 modern and 51 ancient males), along with newly collected and excavated material (77 modern and 23 ancient males), have allowed us to focus on the Y-chromosome lineages identified in the ancient and the modern data and propose a combined study of paleogenetic, historical and archaeological data, confronting different approaches in order to shed light on the evolution of cultural practices and social structure in the Yakut population during the last five centuries.

The genetic structure of ancient populations can shed light on their social structure and their cultural practices[11]. Previous studies have shown that the comparison of Y-chromosome lineage diversity and mitochondrial lineage diversity could allow the identification of pastoralist and farmer societies[12]. Male lineage diversity can in fact reveal clan or lineage exogamy and is therefore an indication of matrimonial structure.

Previous studies of Y-chromosome lineages have demonstrated a strong link between the Yakuts and Turkic South Siberian populations[13], consistent with the historical, archaeological and linguistic consensus, as well as mitochondrial lineage data[14]. While the precise timing of the waves of northward migration remains uncertain, they likely occurred no earlier than the eleventh century and no later than the fifteenth century. The genetic homogeneity of the modern population has also been described as a consequence of a very strong bottleneck event associated with that northward migration[15,16]. Previous studies of archaeological material[5,7] have also shown that genetic diversity in the archaeological population was low and that there were few differences in the nature of Y-chromosome lineages between ancient and modern individuals.

The present study explores the demographic events that took place after the establishment of these Turkic populations in what is now called Central Yakutia. It has been evident from the first studies that this large collection of Yakut graves, dated between the seventeenth century and the nineteenth century, was not representative of the ancient population. It is not a set of cemeteries, where the deceased might be routinely buried, since such practices were almost inexistent in medieval Yakutia, where burial itself is a difficult and time-consuming task because of the frozen ground. It is rather a series of isolated graves, with some archaeological sites discovered tens of kilometres away from any

other grave. The homogeneity of grave goods does not extend beyond a few very characteristically Yakut elements, which evolved recognisably throughout the period and adapted to different biomes. Most graves present one or more original elements that make finding a common classifier a difficult task. Since mitochondrial diversity seems to be maintained throughout the period in all regions[6], we endeavoured to determine whether Y-chromosome lineage diversity could be linked to the selection of individuals for burial, rather than perishable disposal, as was usual. We therefore identified lineages and placed them in a chronological and geographical context to better understand how the paternal lineages represented in the archaeological record conferred burial privileges to the individuals who carried them.

Having identified favoured lineages, we focused on the graves that were associated with them to determine whether they contained other signs of high social status or reasons for preferential burial. While some burials could be explained by epidemiological or religious factors (shamanic or Christian), the presence of Russian-made artifacts led us to focus on a few graves, especially the grave of a woman buried on the known territory of a powerful medieval clan. This gave us the opportunity to explore historical and archival data associated with the Russian conquest of Siberia that seemed to directly link the rise to power of a local tribe (the Kangalaszy) to that specific grave, while genetic data indicated a link between the occupant of the grave and the dominant Y-chromosome lineage.

As in all societies, Yakut demographic history has been placed within a mythological framework[17], with named characters, most notably from the Kangalaszy tribe, held responsible for cultural advances, military victories, defeats and the conquest of new territories. Because palaeogenetics have at times helped identify ancient figures, such as ancient monarchs[18], some findings have been directly linked to historical characters, especially military leaders[19]. However, demographic models show that the dominance of some haplotype in a population can be the result of the natural distribution of haplotypes[20]. Furthermore, tribes often describe themselves as descent groups, with all members sharing a common ancestor. While, in some societies, all men of the same clan might share a lineage, tribes are in fact more likely conglomerates of clans bearing different lineages[21]. We endeavoured to determine whether the existence of specific historical and/or legendary Yakut figures was in fact compatible, necessary or helpful in explaining the events that took place in recent centuries.

While individual identification is commonplace in forensic genetics[22], it is usually outside the reach of archaeological genetics. This work presents the rare case of ancient DNA studies leading to the identification of an individual and their place in the recent history of cultural evolution and power changes in the Yakut population of North-Eastern Siberia (now the Sakha Republic).

## Results

**Low Y-chromosome lineage diversity**. We found that most Yakut 17-STR profiles (in total $n = 340$) only differ from each other by <3 mutated positions, across all epochs and locations (Supplementary Information 1 and Supplementary Fig. 1). Furthermore, several haplotypes are represented by more than one individual. This observation is only partially explained by the presence of related individuals in the graves that were studied. Many male subjects share a paternal line even though they do not share a first-degree relationship[6].

Among the 74 ancient males, there were only 21 unique Y-STR haplotypes, 12 of which were carried by isolated individuals, the remaining 9 being shared by 2–36 individuals. Male lineage

**Table 1 Notable Y-chromosome haplotypes.**

| | DYS456 | DYS389 I | DYS390 | DYS389 II | DYS458 | DYS19 | DYS385a | DYS385b | DYS393 | DYS391 | DYS439 | DYS635 | DYS392 | GATA_H4 | DYS437 | DYS438 | DYS448 |
|---|---|---|---|---|---|---|---|---|---|---|---|---|---|---|---|---|---|
| Ht1 (dominant) | 14 | 14 | 23 | 32 | 16 | 14 | 11 | 13 | 14 | 11 | 10 | 22 | 16 | 12 | 14 | 11 | 19 |
| Ht2 (central) | 14 | 14 | 23 | 31 | 16 | 14 | 11 | 13 | 14 | 11 | 10 | 22 | 15 | 12 | 14 | 11 | 19 |
| Ht3 (western) | 14 | 14 | 23 | 31 | 16 | 14 | 11 | 13 | 14 | 11 | 10 | 22 | 16 | 12 | 14 | 11 | 19 |

diversity is therefore very low in the ancient sample (but this is expected given non-random burials). The 266 modern males carried 106 different haplotypes, 82 of which were unique and 34 of which were shared by 2–73 individuals. While modern diversity is significantly higher than ancient diversity, it is very low compared to other populations[23], with a Haplotype Diversity of 0.902256[24] and only 30.8% (82/266) of individuals carrying unique haplotypes.

Three haplotypes dominate the modern population (Table 1): Ht1, carried by the majority of men; Ht2, characteristic of Central Yakutia; Ht3, dominant in Western Yakutia.

Overall, modern and ancient Y-chromosome haplotypes are either identical or closely related and the composition of the modern living male population shows many similarities with the archaeological samples, although they necessarily only provide a partial picture of the ancient Yakut population.

**Haplogroups and comparable lineages in other populations**. Among the 21 different haplotypes identified in the ancient Yakut population, 17 belonged to the N1a1-M46 haplogroup (92% of individuals), 1 belonged to the N1a2-CTS6380 haplogroup (2 individuals), 1 to the C2-M217 haplogroup (2 individuals) and 1 to the C2b1a1b1-F3985 haplogroup (1 individual). Finally, one individual could not be affiliated reliably (Supplementary Data 1).

Searching for matching data in an in-house database containing more than 200,000 Y-STR haplotypes revealed that all 17 different N1a1-M46 haplotypes can be found (over 85% match, including 7 exact—100%—matches) in the modern Yakut population. Among those, two are also found (including exact matches) in the modern Buryat population of southern Siberia. One of these two Buryat haplotypes belonged to an undated individual but the second one belonged to an individual buried before the seventeenth century in Central Yakutia. Results in ancient data (matches over 70%) were composed of diverse Mongol, Turkic and Yakut individuals.

The N1a2-CTS6380 haplotype, shared by two individuals anterior to the seventeenth century from Central Yakutia and the Vilyuy (western) region, was found (including 100% matches) in modern Khakassians. Results in ancient data were also composed of Mongol, Turkic and Yakut individuals.

The haplotype belonging to haplogroup C2-M217 was carried by two undated individuals buried close together in Central Yakutia. It matches (up to 100%) haplotypes found in the modern Buryat population and more than 100 modern Mongols. No ancient matches were found.

The individual belonging to haplogroup C2b1a1b1-F3985 was buried in Verkhoyansk before the seventeenth century. His haplotype was not found in any modern population but six matches over 70% were identified in ancient individuals from the European Steppe, possibly indicating an Altaic origin for this extinct lineage.

The one haplotype that could not be affiliated reliably (Musée ethno) was also undated and therefore not included in further analyses (no undated samples were). A matching haplotype (71%) was, however, found in an Early Okunevo individual. Since this analysis was performed on an individual presented in a museum with limited identification, these results could suggest a much earlier date for the skeleton (likely the Bronze Age).

**The geographical dispersion of male lineages**. We analysed the geographical distribution of archaeological remains and modern individuals in relation to their Y-haplotype (Fig. 1). We distinguished four regions within Yakutia: Central Yakutia (around the capital, Yakutsk), Vilyuy (West, in the valley of the Vilyuy

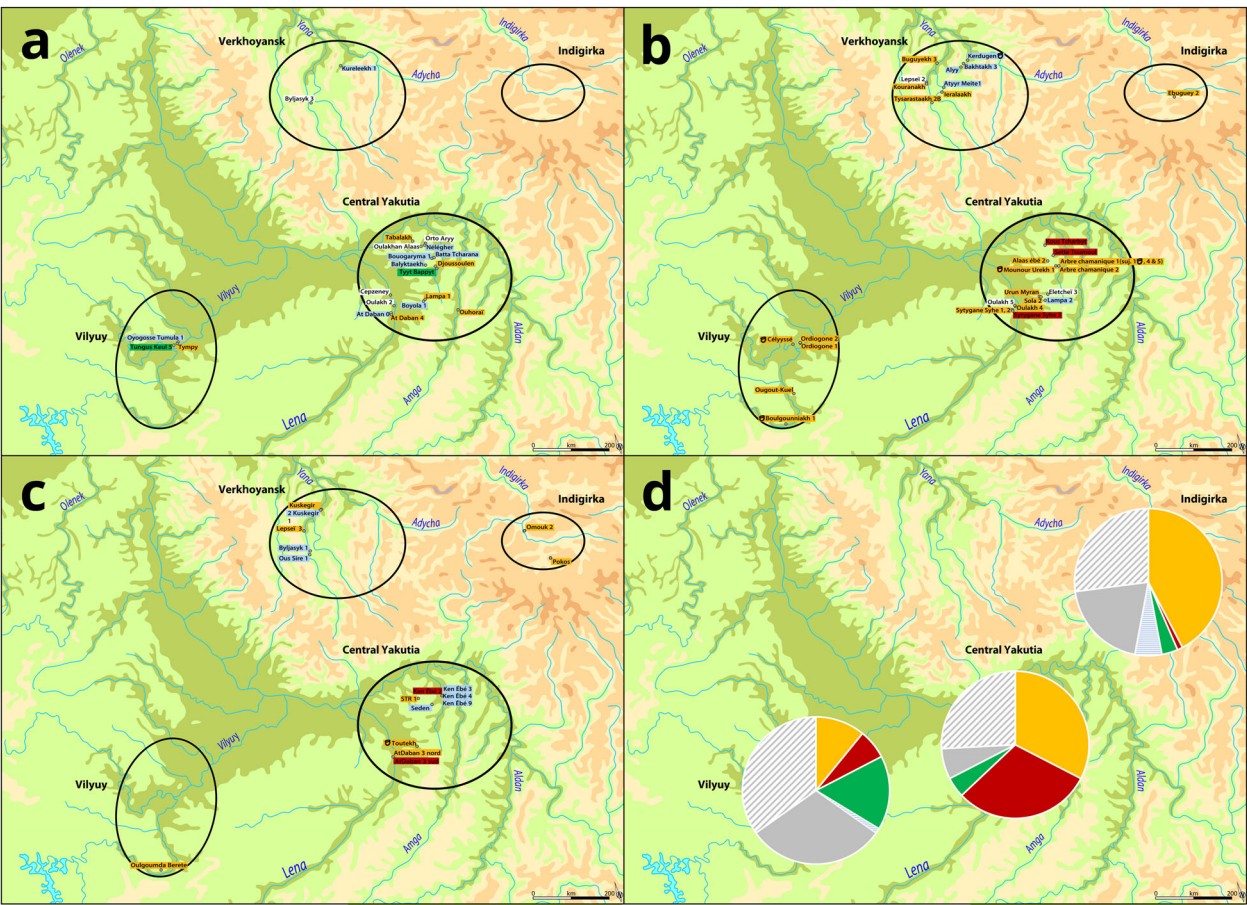

**Fig. 1 Geographical distribution of Y-haplotypes through time. a** before 1700; *n* = 21, **b** eighteenth century; *n* = 32, **c** nineteenth century; *n* = 17, **d** modern haplotypes; *n* = 266; yellow: Ht1 individuals, red: Ht2 individuals, green: Ht3 individuals, blue and horizontal blue stripes: minor haplotypes also found in the archaeological record, grey: haplotypes only found in modern samples, grey oblique stripes: unique haplotypes, white: unique haplotypes. All names refer to one individual in a single grave (e.g., "Ordiogone 1" is the first grave/individual excavated at the Ordiogone site). Two exceptions are Sytygane Syhe 1 and 2, juxtaposed graves, and Arbre Chamanique 1 (suj.1, 4 and 5), three individuals buried in a quintuple grave (other individuals were female).

River), Indigirka (North-East, in the valley of the Indigirka River) and Verkhoyansk (North, isolated within a mountain range).

Before the year 1700 (Fig. 1a), both the Ht1 (dominant) line and the Ht3 (western) line were already present in Central and Western (Vilyuy) Yakutia. Other (minor) lineages (Nelegher, Boyola 1, Kureleekh 1/At Daban 0) have survived until the present day, while others (Balyktaekh/Oyogosse Tumula 1, Batta Tcharana/Bouogaryma 1) have not.

The eighteenth century (Fig. 1b) is marked by the appearance of the Ht2 line in the archaeological record (exclusively in Central Yakutia) and the expansion of the Ht1 (dominant) line into the regions of Verkhoyansk and the Indigirka. It is also the period in which the Ht1 line is most dominant. The minor lineage of Kureleekh 1 is still represented (by Lampa 2, Atyyr Meite 1 and Alyy), but the only other eighteenth century non-unique minor lineage (Kerdugen and Bakhtakh 3) has not been found in modern samples.

The Ht1 line remained dominant in the nineteenth century (Fig. 1c), when it could be found in all four regions. The Ht2 line was again only found in Central Yakutia. While minor haplotypes are more numerous, only one (carried by Byljasyk 1 and Ous Sire 1, the line of Boyola 1) has survived into modern times.

Modern data collected in three of the four archaeological regions (Fig. 1d) show the sustained dominance of the Ht1 line in Central (29/89 men, 33%) and Northern Yakutia (36/85 men, 42%), while the Ht3 (western) line is the most frequent in the

Vilyuy region (15/92 men, 16%). In Central Yakutia, the Ht2 (central) line is the second most represented, with numbers comparable to the Ht1 (dominant) line (27/89 men, 30%). Additionally, it should be noted that the Ht1 (dominant) line was the only one found in archaeological samples from the Indigirka region, that the Ht2 (central) line was only found in archaeological samples from Central Yakutia and that 68% (15/22) modern carriers of the Ht3 (western) line live in the Vilyuy region. Therefore, while all three dominant lineages are found in all regions, they each characterise one of these regions.

Most men descended from surviving ancient minor lineages were found in the Indigirka region. They belonged to three such lineages. Finally, 50% (132/266) of the participants carried haplotypes that were not found in the archaeological record. Almost none of these men carried non-Yakut (Ukrainian, Russian, Kazakh, etc.) haplotypes. In fact, the great majority of unique or exclusively modern haplotypes show very few differences with the dominant Yakut haplotypes (Supplementary Information 1 and Supplementary Fig. 1).

**The dominance of a single male line (Ht1) in the eighteenth century.** Haplotype distribution across four dated periods (before 1700, 1700–1800, 1800–1900 and after 2000) shows the continued dominance of the Ht1 haplotype (Fig. 2), even in modern data. At any given time, more than 25% of Yakut men carry the Ht1 Y-chromosome lineage. In eighteenth century samples, it

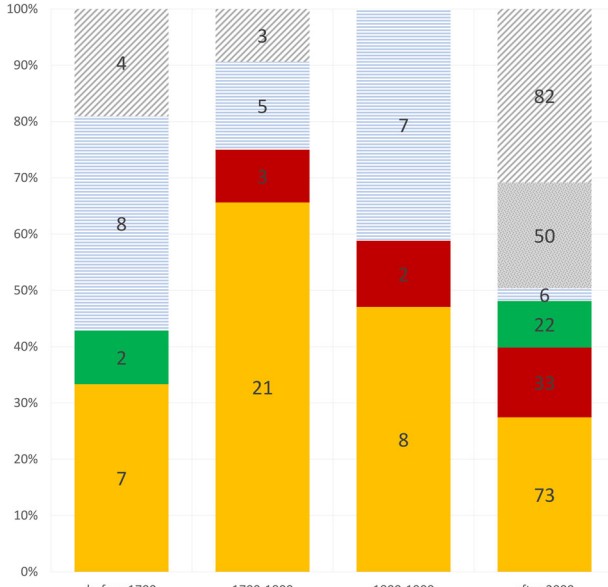

**Fig. 2 Haplotype distribution in the four periods.** Yellow: Ht1 individuals, red: Ht2 individuals, green: Ht3 individuals, blue horizontal stripes: minor haplotypes also found in the archaeological record, spotted grey: haplotypes only found in modern samples, grey oblique stripes: unique haplotypes.

constitutes the absolute majority (more than 60%) of archaeological data.

The Ht2 (central) lineage was not found before the eighteenth century, but it afterwards constituted the second most common haplotype in all Yakut male samples. While the Ht3 (western) line was found in two samples dated before the eighteenth century, it does not reappear in other ancient samples, even though it is the third most common lineage in modern samples. Other minor ancient lines were found in different periods and three of them have survived to the present day. None of them were found in more than three individuals in any single period.

Some haplotypes were unique (belonging to only one individual in one period). Such haplotypes were more frequent in the archaeological record before the eighteenth century and in modern samples. Finally, 50 haplotypes belong to two or several unrelated modern individuals and had not been found in archaeological samples.

Given pre-eighteenth century census estimations[17], we constructed a model of the demographic expansion of the Yakut population between the early eighteenth and the twenty-first century. We were therefore able to determine whether the dominance of the Ht1 line during the 1700–1900 period was consistent with the normal evolution of the population, or if it constituted an anomaly.

Tests show (Table 2) that haplotype diversity in the eighteenth century and in the nineteenth century is expected to be significantly higher than the observed values. Therefore, the dominance of the Ht1 line (and possibly the Ht2, "central", line) in the archaeological record is anomalous and cannot be the result of random burials within a natural demographic history.

**A notable member of the Ht1 clan buried in the At Daban 6 grave.** According to Yakut historical records, Mazary Bozekov was a Yakut *toyon* (chieftain), grandson of Tygyn Darkhan, himself a renowned seventeenth century *toyon*[25]. He had six sons and died in 1703, at age 75[17]. He was buried in the Erkeeny valley on a terrace then called Tuekei and now Istiakh, known to be a

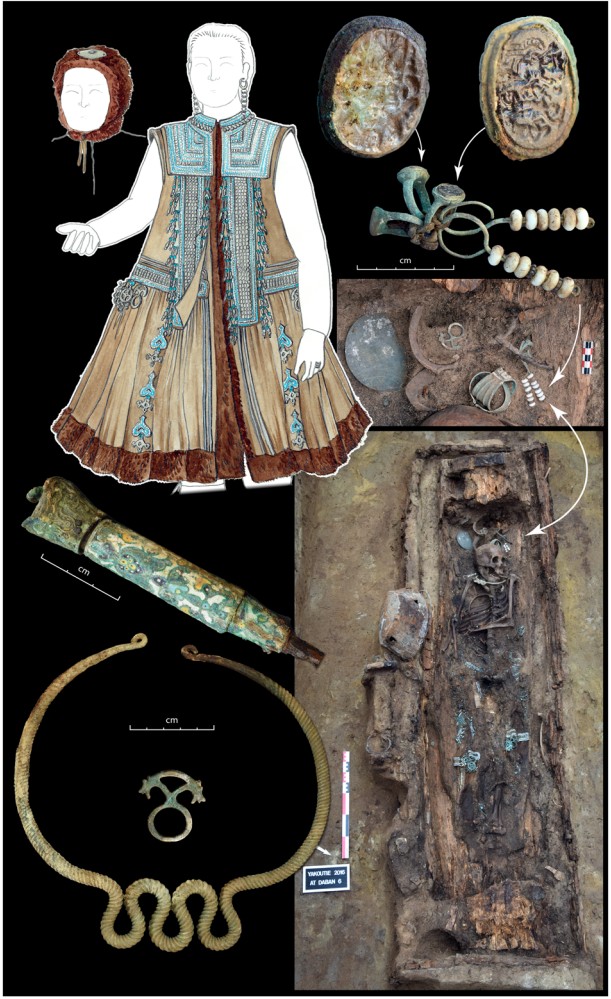

**Fig. 3 The At Daban 6 grave.** The knife-handle made of lobed enamel is displayed, as well as the signet rings. The reconstruction of the dress includes a very large number of imported glass beads, which were recovered during excavation.

cemetery for Tygyn's descendants[26]. His grave was excavated in August 1933 by the Director of the Yakutsk Museum, M.A. Kovinin, and researcher, G.V. Ksenofontov. In the grave, the skeletal remains of a man were discovered. The grave goods consisted of an iron cauldron, a leather container for Kumis, a saddle and a horse bit, three buttons and cufflinks with traces of encrusted enamel, the metallic parts of a belt, a bow and a quiver with 18 arrowheads of iron and bone. The subject wore two rings, one on each hand, one of which had inlaid stone[26]. While no skeletal elements were collected from the grave of Bozekov and therefore no genetic analyses performed, the notes taken by researchers at the time seem to indicate that the outline of the burial could still be found somewhere on Istiakh.

In 2016, we conducted an archaeological excavation south of Yakutsk, on the At Daban site, 2 km from the terrace called Istiakh where Mazary Bozekov had been buried. Among the graves that were excavated was the grave of a woman, At Daban 6, very richly furnished (Fig. 3). Two items were especially notable: (1) two Russian signet rings attached to one of her earrings in the offerings chest, as if left by a family member who added them to the grave goods (Supplementary Information 2 and Supplementary Figs. 2 and 3); (2) two buttons of lobed enamel and a dagger with a lobed enamel handle from Velikiy Ustiug (Western Russia)[27]. The enamel of these two items was

**Table 2 Numbers of different haplotypes in the four periods.**

| Period | Before 1700 | 1700–1800 | 1800–1900 | After 2000 |
|---|---|---|---|---|
| Sample size | 21 | 32 | 17 | 266 |
| Modelled number of haplotypes | 11.65 | 15.44 | 11.60 | 108.83 |
| Observed number of haplotypes | 11 | 7 | 6 | 106 |
| $p$ value | 0.477 | 0.002[a] | 0.008[a] | 0.413 |

[a]Observed number of haplotypes is significantly inferior to modelled number of haplotypes.

manufactured in the seventeenth century for the Tsar's court. The only comparable items found in Yakutia are the three buttons and cufflinks with traces of encrusted enamel that were found in the grave of Mazary Bozekov. One of these rings bore a heraldic pattern and the second a hagiographic pattern. Both could be used as seals since the pattern was embossed in the metal[28]. The heraldic ring's escutcheon represents an eagle supported by two affronted lions rampant. The crest is a crowned nobleman's helmet, itself with a crest ornate with three ostrich feathers. These arms are typical of a powerful Russian family, possibly the Kvashnin-Samarin (Supplementary Information 2 and Supplementary Fig. 2). The hagiographic ring represents several characters standing around a king sat in his throne. A character is knelt at the feet of the king, while a soldier holds a sword in one hand and a child by its feet in the other. A woman stands behind the soldier and supports his arm, as if encouraging him to strike. These characters and their attitudes are characteristic of a biblical scene: "The Judgment of Solomon" (Supplementary Information 2 and Supplementary Fig. 3). This scene is an allegory of equitable and wise judgement. Having visited the Tsar, Bozekov acquired the title of Duke and was given the right to arbitrate judicial disputes between Yakut clans[17]. In the context of that period, this would have provided him with a ring for each of the two titles, gifts from the Tsar or from a noble family, a Judge and a Duke. The two signet rings found at At Daban 6 could therefore be those of Bozekov, deposited in the grave of one of his relatives. All results are therefore consistent with At Daban 6 having been a relative of Mazary Bozekov.

Genetic analyses have come to complete our understanding of the At Daban 6 grave. Autosomal and mitochondrial analyses show that she was the mother of an individual buried 2.5 km from her, at the Sytygane Syhé 1 site (SS1): an adult male who died between 1700 and 1750 (Supplementary Information 3 and Supplementary Data 2). The SS1 individual was a member of the Ht1 (dominant) line (Fig. 1b). This would indicate that Bozekov, as well as his paternal grandfather Tygyn Darkhan, could have been members of the Ht1 (dominant) line.

**The expansion of Ht1 is the expansion of the Kangalaszy.** The Russian colonisation of Yakutia began during the first half of the seventeenth century[17] and the period was marked by conflict between the Cossacks, serving the Tsar, and the Kangalaszy, then the most powerful of Yakut tribes. The subsequent Russian victory and colonial peace were followed by the rapid expansion, in the last years of the seventeenth century, of a few Yakut clans into territories then occupied by autochthonous Tungus, Yukaghir and other tribes. This Yakut "Golden Age" shaped ethnic composition, land use and cultural practices in what is now the largest of all Russian republics.

While colonial war against the Cossacks is associated with the name of Tygyn Darkhan, defeated *toyon* (chieftain) of the Kangalaszy and his son Bozeko, executed by the Russians in 1642[29], the later prosperous period is associated with the name of one the grandsons of Tygyn, the son of Bozeko, Mazary Bozekov, also *toyon* of the Kangalaszy, one of the first Yakuts to visit Tsar

Feodor III in Moscow[17,25,30,31]. We have shown that known members of the Kangalaszy were associated with the grave of At Daban 6 and the Ht1 line, especially those buried in the historical territory of that clan.

These results therefore suggest that the expansion and dominance of the Ht1 haplotype are the direct result of the consolidation of power around the Kangalaszy clan, from relative confinement in Central Yakutia and the Vilyuy in the seventeenth century, to distant northern and north-eastern regions, as early as the eighteenth century. This expansion is especially visible in the population of the Indigirka, which was composed of autochthonous Tungus tribes before the eighteenth century and is now dominated not only by Yakuts, but the Ht1 lineage specifically. In effect, centuries after the bottleneck event at the foundation of the Yakut ethnic group (which is supported by the data presented here), male haplotypes underwent another bottleneck event caused by the conflict between different lineage groups[32].

The Ht3 (western) line was already present before the Russian period but none of its carriers were buried during the 1700–1900 period, even in the Vilyuy region, where this lineage is dominant today. Three scenarios could account for this anomaly: (1) this line took hold in the Vilyuy before the Russian period and was in fact not widespread in Central Yakutia, (2) the Ht3 line was a lineage from Central Yakutia that was driven out in the eighteenth century or (3) the Ht3 line only came to prominence in recent times. In any case, the Ht3 (western) line appears to have been denied burials during the period of the dominance of the Ht1 line, either because its carriers had fewer contacts with Russian culture, or because burials were a privilege they were denied.

**The preferential burial of the elite.** The burial of corpses was uncommon before the Russian era in Yakutia. The practical difficulties (permafrost) and the absence of religious injunctions (before Christianisation) most often caused deceased individuals to be deposited in suspended wooden coffins called *arangas* (Supplementary Information 4 and Supplementary Fig. 4), very few of which have survived to this day. The remains associated with these *arangas* were usually entirely degraded in the years following the funeral. Shamans were sometimes buried after that initial stage of degradation and the victims of epidemics were usually left in their homes. Despite this, both primary shamanic graves[33] and the graves victims of epidemics have been uncovered[8,9,34]. Early Yakut Christians were also buried, even before the Christianisation of most of the population in the nineteenth century. Other graves, moreover, do not fall in any of these three categories. Some individuals were buried with traditional Yakut grave goods, no Christian crosses or shamanic objects and bearing no signs of disease.

We have shown that Y-haplotypic diversity in eighteenth century and nineteenth century graves was significantly lower than expected given the diversity in the more ancient and the modern population. This suggests that male burials were selective, especially reserved to members of the Ht1 (dominant) line. It is an indication of the cultural influence of Russian invaders on

some clans after the beginning of colonisation[30]. It is especially notable that the Ht3 (western) line, already present before the year 1700 and dominant in the Vilyuy region today, was not found in any eighteenth or nineteenth century grave. Conversely, carriers of the Ht2 (central) line have been buried in Central Yakutia during the domination of the Ht1 haplotype.

**Mythological explanations and their limitations.** A previous study by Tikhonov et al.[19] has proposed a combination of Y-STR analysis and genealogical reconstruction to attach historical and legendary names to haplotypes Ht1 and Ht2. The first would belong to Elley Botur and the second to Omogoy, two characters of legend, said to have introduced Yakut culture in the valley of the Lena, some few centuries before the period described here[1,35]. The data presented here rely on the analysis of additional markers and the associated archaeological material, but it also indicates the dominance of two lineages in Central Yakutia (the valley of the Lena). We have also shown that a third haplotype, already underlined by Tikhonov and his team (Ht3), is dominant in the western regions of Yakutia. Since Yakut ethnography suggests the existence of three Yakut founding ancestors (*us Sakha*), this haplotype could be attached to Uluu Khoro, Omogoy's brother-in-law[36], whose name is characteristic of western Yakut groups.

Setting aside legendary or semi-legendary characters such as Elley or Omogoy, three male lineages appear to have dominated pre-colonial Yakutia. As shown by historical data, the clan of Tygyn, already the most powerful, encountered the Russian invader and, although it was defeated militarily, was able to use a privileged position to assert dominance over other Yakut clans around its ancestral territory and Tungus tribes in the far North.

Although legendary accounts vary, it is told that Elley supplanted Omogoy in the distant past, extending his reach and that of his sons over all of Yakutia. It has therefore been proposed that the legend of Elley, in its modern form, could be a retelling of the expansion of the line of Tygyn in the early eighteenth century, after the start of Russian colonisation[37]. This is compatible with the legendary notion that Tygyn was the grandson of Elley himself[38] and that he fought the descendants of Omogoy (including the western clans of Vilyuy, Nam, Bayagantay and Khoro)[36].

Some mythological or mythologised narratives fit the succession of historical events but are not supported by paleogenetics in-and-of themselves[20]. In the present case, however, the grave at At Daban 6, and its ties with the previously excavated grave of Mazary Bozekov, supports the existence of a link between the lineage of Tygyn Darkhan and haplotype Ht1. The cases of Omogoy and Uluu Khoro are not directly supported by such evidence, but they can be understood within the geographical distribution of graves and modern haplotypes. Omogoy could be a legendary name attached to male clans which included the Ht2 haplotype, defeated by the clan of Tygyn before Russian colonisation. The character Uluu Khoro could have represented the head of a western clan or tribe (which included the Ht3 haplotype), also in conflict with Tygyn Darkhan and denied burials during the domination of his sons and grandsons.

## Discussion

We show that, in the case of the ancient Yakut population, genetic lineage composition is a direct indication of historical events. The foundational bottleneck at the origin of the Yakuts was followed by a second event, a few centuries later, especially observed in the Yakut colonisation of northern regions. Some male clans attained a position of social dominance that increased the frequency of their Y-chromosome lineages in the population.

Discrepancies between archaeological populations and their modern counterparts are not uncommon, but they can rarely be quantified. Our study shows that there was considerable bias in selecting Yakut men for burial in the eighteenth century. Some funeral practices were reserved to a specific group; in this case likely one or two dominant male clans or lineages, including the lineage that dominated the colonisation of the North.

Although the grave of At Daban 6 did not bear a name, the woman occupying is not only an anonymous member of the ancient Yakut population; it is possible to determine her place within the large-scale transformation of her culture in the eighteenth century. This is another suggestion that the diversity of graves in many single-culture archaeological sites is, at least in part, the result of the personal histories of ancient individuals, some of which can be rediscovered.

While the victory of Russian colonisers is undeniable, both in its military and economic aspects (defeated Yakut clans subsequently paid heavy tribute and tax), the colonisation of Yakutia was not a simple process of population replacement. Yakuts still constitute half of the population of the Sakha Republic and both their language and their culture (horse and cattle breeding, as opposed to reindeer-herding and hunting) are dominant over all the territory.

Cooperation between historians, archaeologists, biologists and other specialists has allowed a better understanding of the recent History of the populations of North-Eastern Siberia. Many graves remain undiscovered in the vast expanse of North-Eastern Siberia and new genomic approaches will surely yield more evidence and contribute to our understanding of the genetics of colonisation in Yakutia.

## Methods

**Samples and haplotype comparisons.** We selected data from previous studies that included the 17-STR loci from the AmpFLSTR Y-Filer kit so it would be possible to perform all comparisons and build models using those 17 Y-STR positions. Because studies by our team and the teams associated sometimes produced new data from the same human remains, either to improve the quality of allele determinations or to increase the number of Y-STR markers, the lists of archaeological samples from each study often overlap (Table 3). The present study includes 23 unpublished ancient samples and 77 unpublished modern samples, collected during a dedicated expedition[39]. The method of repartition into chronocultural phases is shown in Supplementary Information 5, along with examples of the material found in those phases (Supplementary Information 5 and Supplementary Figs. 5–9). All Y-STR haplotypes are collected in Supplementary Data 1. Haplogroups were assigned when possible using Nevgen Y-DNA haplogroup predictor (www.nevgen.org). The haplotypes themselves were compared with an in-house database of more than 200,000 Y-STR haplotypes, including some from ancient material. For each haplotype, we retrieved modern haplotypes that showed more than 85% of matching alleles and ancient haplotypes that showed more than 70%.

Tests were implemented on 74 ancient samples and 266 modern samples (Supplementary Data 1). This study was approved by the local Committee on Biomedical Ethics of the Yakut Scientific Centre of Complex Medical Problems. Informed consent was obtained from volunteers, or the parents of participating children where applicable, before sample collection. All simple comparisons and descriptive statistics were computed in R[40]. Median-joining Network

**Table 3 Compiled samples.**

| Type of samples | Number of samples | Number of Y-STR loci | References |
|---|---|---|---|
| Ancient | 27 | 17 | Crubézy et al.[7] |
| Modern | 133 | 17 | Thèves et al.[10] |
| Modern | 21 | 24 | Gao et al.[49] |
| Ancient | 39 | 17, 24 | Keyser et al.[6] |
| Modern | 35 | 24 | Zvénigorosky et al.[4] |
| Ancient | 23 | 17, 24 | This study |
| Modern | 77 | 24 | This study |

All 17 Y-STR haplotypes (n = 340) can be found in Supplementary Data 1.

representations were produced using the Network software (fluxus-engineering. com)[41] before they were simplified.

**Archaeological material**. The sample set presented here was assembled during 17 years of archaeological digs in several regions of the Sakha Republic[3–10] by the French Archaeological Mission in North-Eastern Siberia (MAFSO). Prospective campaigns identified potentially interesting sites in isolated locations, most often small hills on the outskirts of forests (in the many meadows that make up the Taiga landscape of Yakutia). On some sites, several graves were excavated, and some contained more than one individual. An overview of characteristic grave goods and chronological phases is presented in Supplementary Information 5.

We present 23 ancient males and 1 female, excavated by the MAFSO, that had not yet been described in the literature. The latter is At Daban 6, presented in Fig. 3. The location and dating of all 23 male individuals can be found in Supplementary Data 1 (classified as "this study"). They were discovered in all four regions under study (Central Yakutia, the Vilyuy Region, the Indigirka Region and the Verkhoyansk Region) and distributed among all three ancient periods presented here (before 1700, eighteenth century, nineteenth century).

**DNA typing and sequencing**. All ancient samples were collected by the MAFSO, every summer since 2002. Extraction and amplification protocols for modern and ancient samples have been described in Ricaut et al.[5], Crubézy et al.[7], Thèves et al.[10], Biagini et al.[8], Keyser et al.[6] and Zvénigorosky et al.[4]. Allele calls were considered reliable if they had been unambiguous in at least two amplifications from two different DNA extracts. Given the repetition of analyses throughout the years with the improvement of technology, most samples have been subjected to many more amplifications and extractions. Consensus STR haplotypes and genotypes have been updated whenever necessary.

All previously unpublished male samples (23 ancient and 77 modern) were analysed at 24 Y-chromosomal STR loci (DYS19, DYS385a/b, DYS389I/II, DYS390, DYS391, DYS392, DYS393, DYS437, DYS438, DYS439, DYS448, DYS456, DYS458, DYS635, Y GATA, DYS449, DYS460, DYS481, DYS518, DYS533, DYS570, DYS627 and DYF387S1a/b) using the AmpFLSTR Y-Filer Plus kit (Life Technologies[TM]). All STR products were run on the 3100 or 3500 genetic analysers (Life Technologies[TM]) and analysed using GeneMapper v. 4.1 (Life Technologies[TM]).

Additionally, all archaeological samples from the MAFSO were analysed at 15 or 21 autosomal STR loci (using the Identifiler and GlobalFiler kits from Life Technologies[TM]), some published alongside Y-STR data and some remaining unpublished. These data were used for the determination of genetic kinship between individuals or graves[6]. The present study refers to one previously unpublished genetic relationship and therefore includes the two relevant autosomal STR genotypes and the set of allelic frequencies (Supplementary Data 2) used to implement kinship testing via the likelihood ratio method, performed with the Familias package for R[40,42–44].

Mitochondrial HV1 haplotypes for the two individuals whose kinship is discussed are also presented, as a confirmation of their familial relationship (Supplementary Data 2). One was produced in a previous study[5] and the second was analysed for this study. Three hundred and eighty-one base pairs of the HVI region of the mtDNA genome were amplified and sequenced in two overlapping fragments, as described in previous publications[45,46].

Five ancient samples were subjected to whole mitochondrial genome sequencing as a check of the quality of DNA extracts. The analysis was performed on the Ion Torrent Personal Genome Machine[TM] (Thermo Fisher Scientific) using the Precision ID mtDNA Whole Genome Panel (Applied Biosystems). DNA libraries were constructed using the Ion Ampliseq[TM] Library kit (Thermo Fisher Scientific) with 2 separate primer pools amplifying 162 amplicons that target the entire human mitochondrial genome. These analyses, that confirm the high quality of the material recovered, are presented in Supplementary Data 3.

**Demographic model and coalescent analysis of Y-haplotypes**. We performed a demographic simulation of male lineage composition using the BayeSSC programme[47,48], in order to determine if sets of ancient samples could correspond to an ancient population that naturally evolved into the modern population. The estimated parameters were between 100,000 and 150,000 adult males in the modern population and between 10,000 and 15,000 adult males in 1700. The probability of one allele on the 17 Y-STR markers (available for all subjects) to mutate by one step was set to 0.003 per generation (the mean mutation rate for the Y-STR markers used), which is a global mutation rate of 0.051 per generation for whole haplotypes.

We ran the model 1,000,000 times and computed the *p* value of an exclusion of the observed value on one side (i.e., we considered the null hypothesis that observed numbers of private haplotypes should be consistent with the model and the alternative hypothesis that observed numbers of private haplotypes were smaller than expected in the model, we did not envisage an observed number of private haplotypes greater than expected in the model).

**Reporting summary**. Further information on research design is available in the Nature Research Reporting Summary linked to this article.

## Data availability

All data (Y-STR haplotypes, autosomal STR profiles, autosomal STR frequencies) are provided in Supplementary files. Any data that were not used in the present study but relate to the same archaeological sites are available upon request.

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

## Acknowledgements

Administrative and research work were supported by the programme of the France-Russia Associated International Laboratory (LIA COSIE number 1029), associating the North-Eastern Federal University (Yakutsk, Sakha Republic), the State Medical University of Krasnoyarsk, the Russian Foundation for Fundamental Research (Moscow, Russia), the University of Paul Sabatier Toulouse III, the University of Strasbourg I (France) and the National Centre for Scientific Research (Paris, France). Funding for excavations was provided by the French Archaeological Mission in Oriental Siberia (Ministry of Foreign and European Affairs, France), the North-Eastern Federal University (Yakutsk, Sakha Republic), the Institute of Humanities and Issues of the Minority Peoples of the North (Siberian Branch of the Russian Academy of Sciences) and the French Polar Institute Paul Emile Victor. The study was supported by the State Assignment for Fundamental Scientific Research #2019-1472, RFBR grants 18-05-60035 (Arctica) and 19-34-60023 (Perspektiva).

## Author contributions

V.Z.: writing (genetics), revision, collection of modern samples, analysis of genetic data. S.D.: writing (archaeology), collection of ancient samples, analysis of archaeological material. L.R.: writing (history), collection of ancient samples, analysis of historical and archival data. P.G.: production of figures, collection of ancient samples, analysis of archaeological material. C.P.: illustrations, analysis of archaeological material. M.P.: analysis of archaeological material. A.A.: supervision of archaeological operations (Russian side). O.M.: administrative support (Russian side). A.G.: laboratory work, analysis of genetic data. J.L.F.: laboratory work, analysis of genetic data. A.S.: collection of modern samples. G.R.: collection of modern samples. N.B.: administrative support (Russian side), grant proposals. S.F.: supervision of modern sample collection (Russian side). B.L.: supervision of modern sample collection (French side). E.C.: supervision of archaeological operations (French side), collection of ancient samples, analysis of archaeological material, analysis of historical data. C.K.: project supervisor.

## Competing interests

The authors declare no competing interests.
