## [Peer Review File · Communications Biology]

Reviewers' comments:

Reviewer #1 (Remarks to the Author):

The investigation is carried out following standard methods in the field. While the research question is intriguing and data seem to be technically sound, the implications of the results seem to be narrow in scope and specific to that regional context. Specific Y-lineages are not given the context of how they compare to diversity in other populations. More discussion in relation to previously published research that the authors cite but do not elaborate on, might enhance the paper. Based on its current form, it is not apparent whether the results of this research merits interest from a broader scientific audience. I would suggest that this manuscript is more appropriate for discipline-specific journals such as *Journal of Archaeological Science* or *Archaeological and Anthropological Sciences*.

Reviewer #2 (Remarks to the Author):

The major claims of the paper are that a dominant, immigrant male lineage can be discerned in 18th century Yakuti graves on the basis of a Y-haplotype not observed before the 18th century but which dominated 18th century male burials, with decreasing but still prominent representation in the following centuries. Reference to Yakuti history links this dominant Y-haplotype to the powerful Tygyn lineage.

To the best of my knowledge, these claims are novel, and would be of interest to a wider field in demonstrating the capability of DNS analysis for confirming and contextualizing local history.

I could find no technical, conceptual or other flaws that would obstruct publication. On the contrary, the paper is written well and economically, the content is interesting and the scientific and statistical documentation is convincing.

Reviewer #3 (Remarks to the Author):

The study „The genetic legacy of legendary and historical Siberian chieftains“ summarizes Y chromosomal STR profiles of 74 ancient and 266 modern individuals from North-East Siberia. The Yakuts are one of the genetically best studied Siberian peoples, published reports are numerous on uniparental markers of different ancient and modern Yakut populations. The team of this paper already analysed “150 ancient individuals (15th-19th century) and 510 modern individuals” as they claim in the abstract (I don't know the authorlist of the present paper).

The results, which the authors use in reaching their conclusions have been mostly published already, novel samples coming to this study from 23 ancient and 77 modern individuals, as it turns out from the Material and Methods part of the paper.

The study identifies and follows dominant and minor Y chromosomal lineages though time in Yakutia. Some of them were already prevalent before 1700, others show regional pattern or has not been detected yet from the period between 18-20th centuries. The paper highlights three Y-STR lineages, one of them still dominates present-day Indigirka and Central Yakutia, and connects this lineage (called as “Ht1”) to the chieftain Mazari Bozekov and the Kangalaszky tribe. This connection is considered as proven by a mother-son genetic relation between an unnamed noble women and a male burial with Ht1 Y-STR haplotype 2,5 km apart from his putative mother.

The paper is rich in ethnographic, archaeological and historical details but genetic conclusions are rather lacking. Genetic results appear in the supplement (Supplementary Material 2, 3), and explained in the Material and Methods, mostly technically sound and well described. However, the genetic results are rather shortly mentioned in the main text, they remain in the background. Ethnographic and historical data are well incorporated to the biological clues, but it should not take over the role in the argumentation, and be the proof for the Ht1-Kangalaszky connection e.g. I

think this paper could be interesting not only for historians but also for population geneticists, if it became complemented with genetic overview of the Yakuts population history and a few suggested further analyses.

Major comments, suggestions:

Reading the Introduction part, I missed the summary of the previous genetic analyses conducted on Yakuts (Ricaud et al. 2006, Crubézy et al. 2010, Biagini et al. 2012, Zvéniogorsky et al. 2016), and also the mention of a larger genetic study conducted on Yakuts: Fedorova, S.A., Reidla, M., Metspalu, E. et al. Autosomal and uniparental portraits of the native populations of Sakha (Yakutia): implications for the peopling of Northeast Eurasia. *BMC Evol Biol* 13, 127 (2013) doi:10.1186/1471-2148-13-127

or

Pakendorf B, Morar B, Tarskaia LA, Kayser M, Soodyall H, Rodewald A, Stoneking M: Y-chromosomal evidence for a strong reduction in male population size of Yakuts. *Hum Genet.* 2002, 110: 198-200.

These are all important, and form our knowledge about Yakut population history, although the different methods make previously published data hard to compare.

As it turns out from the Material and Methods, "The present study includes 23 unpublished ancient samples and 77 unpublished modern samples, collected during a dedicated expedition." This information remains hidden in the Abstract-Introduction-Result-Discussion-Conclusion part of the text. I missed from the Material and Methods the authenticity criteria of the ancient DNA results. How many times were these results reproduced? Have mtDNA analyses made the cleanness of the samples and validity of the results a certainty?

The paper neither compare the Y-STR data with other populations of Siberia nor mention Y haplogroups or Y-SNP based subgroup definitions. The study could certainly benefit from a deep Y-SNP test in the N1a (M46) branch.

Line 166: Has the Yakut toyon Mazary Bozekov ever been analysed for Y-STR? That would be a good proof for the lines described in the first part of the Discussion („The expansion of the Ht1 is the expansion of the Kangalszy”).

Lines 196-198: What was the mitochondrial haplotype of the mother and his putative son?

In supplement I only found the followings: "The individuals buried at Sytygane Syhé 1 and At Daban 6 share a mitochondrial HV-1 haplotype and belong 99 to haplogroup D5a2a."

Or is this mitochondrial analysis published already? If not, please describe the analyses in the Methods session and make available the sequence in NCBI/other database or the haplotype in a table. How frequent this haplotype among Yakuts?

The male buried at Sytygane Syhé 1 could be shown in more detail in the supplement. Has he had a rich grave? What kind of gravegoods has he had? It could be presented in the Supplement.

Line 252: Reference to Tikhonov et al. 2019 is missing from the list of references. This study shows quite a few thematic overlap with the current paper under consideration, here own merits should be underlined through the distinction of the two studies.

Why are some of the ancient Yakut graves presented in the supplement but most of them not mentioned? The supplementary table of 340 Y haplotypes include the ancient samples, but I don't see grave numbers or inventory numbers, any specification of these graves. Is Ebuguey 2 or Sola 2 unambiguously identify these specimens?

Minor comments, ordered by occurrence in the main text:

Line 23: „... some of which had already been subjected to genetic analyses“
Please specify, how many?

Line 49: „This work presents the rare case of ancient DNA studies leading to the identification of an individual and their place in the recent history of cultural evolution...“ Their place of an individual? It is confusing.

Lines 59-64: I missed haplotype diversity for ancient individuals. How was the haplotype diversity calculated? Is it the Nei method?

Lines 78-80: Please rephrase the sentence: „Other (minority) lineages...“
I would use minor lineage instead of minority lineage through the text.

Figure 1a-c: highlighted site names are difficult to read, it would be good to read in the figure legends the total number of samples. Figure 1a-c: numbers after site names mean the number of samples? This should be explained in the legend.

Lines 111-112: “while the Ht3 112 (western) line is dominant in the Vilyuy region (15/92 men, 16%).”

It should be rephrased as: Ht3 112 (western) line is the most frequent in the Vilyuy region In Yakutia.

Lines 162: In table 2 the haplotype diversity numbers are very different from those written in Line 63. What is the methodical difference?

Line 186: It would be good to have a larger figure (drawing) about the „The hagiographic ring“ at least in supplement.

Lines 208: Cossacks and the Kangalaszy. It would be good if these two tribe (?) names were introduced earlier in the text, potentially in the Introduction, or detailed in the supplementary material 1 and cited somewhere earlier. In line 208 it is written: „..., then the most powerful Yakut tribes.“ Were Cossack and Kangalaszy those? If yes, please rephrase this sentence (“who were the most powerful..”). If not, please add a sentence about Cossack and Kangalaszy tribes.

Line 224: “In effect, male haplotypes underwent a bottleneck event, caused by the conflict between different lineage groups [Zeng et al. 2018]” The absolute dominancy of the N-Tat Y haplogroups also shows a bottleneck among the Yakuts, a biased reproductive success which is described in previous papers that could be cited more explicitly here.

Line 228: „presence of a lineage in Central Yakutia is anecdotal“ Do you really mean that it was based on personal accounts rather than facts or research? I would write here simply unreliable instead of anecdotal. Or it simply hasn't been detected yet among the 33 studied ancient individuals from this period (state of research).

Reviewers' comments:

Reviewer #1 (Remarks to the Author):

The investigation is carried out following standard methods in the field. While the research question is intriguing and data seem to be technically sound, the implications of the results seem to be narrow in scope and specific to that regional context. Specific Y-lineages are not given the context of how they compare to diversity in other populations. More discussion in relation to previously published research that the authors cite but do not elaborate on, might enhance the paper. Based on its current form, it is not apparent whether the results of this research merits interest from a broader scientific audience. I would suggest that this manuscript is more appropriate for discipline-specific journals such as Journal of Archaeological Science or Archaeological and Anthropological Sciences.

Following this remark and similar remarks from other reviewers, the broader context of the Y-lineages is now presented. We discuss Y-haplogroups, their presence in related populations and especially the known demographic events at the origin of Yakut ethnogenesis. Previously published research, by our team and others, is also now reviewed, in order to give context to the results presented and illustrate how we believe they contribute to our understanding of Yakut demographic history specifically but also colonisation in general.

We hope that, given these additions and modifications, our work might appeal to a broader audience concerned with population genetic diversity and the dynamics of male lineage dominance in general, as well as those anthropological and archaeological aspects that we discuss.

Reviewer #2 (Remarks to the Author):

The major claims of the paper are that a dominant, immigrant male lineage can be discerned in 18th century Yakuti graves on the basis of a Y-haplotype not observed before the 18th century but which dominated 18th century male burials, with decreasing but still prominent representation in the following centuries. Reference to Yakuti history links this dominant Y-haplotype to the powerful Tygyn lineage.

To the best of my knowledge, these claims are novel, and would be of interest to a wider field in demonstrating the capability of DNS analysis for confirming and contextualizing local history.

I could find no technical, conceptual or other flaws that would obstruct publication. On the contrary, the paper is written well and economically, the content is interesting and the scientific and statistical documentation is convincing.

We would like to thank the reviewer for their work and encouraging comments. We have revised the manuscript, adding a review of previous work on the region and discussing the known elements of Yakut demographic history prior to the period discussed in the manuscript.

Reviewer #3 (Remarks to the Author):

The study „The genetic legacy of legendary and historical Siberian chieftains” summarizes Y chromosomal STR profiles of 74 ancient and 266 modern individuals from North-East Siberia. The Yakuts are one of the genetically best studied Siberian peoples, published reports are numerous on uniparental markers of different ancient and modern Yakut populations. The team of this paper already analysed “150 ancient individuals (15th-19th century) and 510 modern individuals” as they claim in the abstract (I don’t know the authorlist of the present paper). The results, which the authors use in reaching their conclusions have been mostly published already, novel samples coming to this study from 23 ancient and 77 modern individuals, as it turns out from the Material and Methods part of the paper.

The study identifies and follows dominant and minor Y chromosomal lineages through time in Yakutia. Some of them were already prevalent before 1700, others show regional pattern or has not been detected yet from the period between 18-20th centuries. The paper highlights three Y-STR lineages, one of them still dominates present-day Indigirka and Central Yakutia, and connects this lineage (called as “Ht1”) to the chieftain Mazari Bozekov and the Kangalaszky tribe. This connection is considered as proven by a mother-son genetic relation between an unnamed noble woman and a male burial with Ht1 Y-STR haplotype 2,5 km apart from his putative mother.

The paper is rich in ethnographic, archaeological and historical details but genetic conclusions are rather lacking. Genetic results appear in the supplement (Supplementary Material 2, 3), and explained in the Material and Methods, mostly technically sound and well described. However, the genetic results are rather shortly mentioned in the main text, they remain in the background. Ethnographic and historical data are well incorporated to the biological clues, but it should not take over the role in the argumentation, and be the proof for the Ht1-Kangalaszky connection e.g. I think this paper could be interesting not only for historians but also for population geneticists, if it became complemented with genetic overview of the Yakuts population history and a few suggested further analyses.

We have reviewed and amended the text as suggested, to provide more information regarding genetic data and conclusions. In particular, we now discuss the place of haplotype Ht1 within the gene pool of Altaic/Turkic populations in Siberia and elsewhere. The following point-by-point answers will indicate where changes and additions have been made.

Major comments, suggestions:

Reading the Introduction part, I missed the summary of the previous genetic analyses conducted on Yakuts (Ricaud et al. 2006, Crubézy et al. 2010, Biagini et al. 2012, Zvéni gorosky et al. 2016), and also the mention of a larger genetic study conducted on Yakuts: Fedorova, S.A., Reidla, M., Metspalu, E. et al. Autosomal and uniparental portraits of the native populations of Sakha (Yakutia): implications for the peopling of Northeast Eurasia. BMC Evol Biol 13, 127 (2013) doi:10.1186/1471-2148-13-127 or Pakendorf B, Morar B, Tarskaia LA, Kayser M, Soodyal H, Rodewald A, Stoneking M: Y-chromosomal evidence for a strong reduction in male population size of Yakuts. Hum Genet. 2002, 110: 198-200.

These are all important, and form our knowledge about Yakut population history, although the different methods make previously published data hard to compare.

We have added a summary of previous genetic analyses performed by our team and discuss the conclusions of Pakendorf's and Fedorova's teams regarding the Yakut gene pool on a larger scale. The following text has been added to the Introduction section:

“Archaeological excavations in the Sakha Republic (Yakutia) in North-Eastern Siberia have recovered the remains of 78 men, 51 of which were subjected to genetic analyses in previous studies focused on biogeographic origins [Ricaud et al. 2004] or kinship between graves [Ricaud et al. 2006, Keyser et al. 2015, Zvéniġorosky et al. 2016] using autosomal STR (Short Tandem Repeat) and mitochondrial (HV1) data. Further studies of biogeographic origins also used Y-STR data [Crubézy et al. 2010] and finally fragments of the genomes of smallpox [Biagini et al. 2012] and tuberculosis [Dabernat et al. 2014] were amplified from some of the same subjects. Anthropological expeditions in Yakutia collected biological samples for more than 200 men, some also subjected to genetic analyses (Y-STR typing) aimed at identifying relationships between the Yakuts and other Siberian populations [Thèves et al. 2010].”

As well as:

“Previous studies of Y-chromosome lineages have demonstrated a strong link between the Yakuts and Turkic South Siberian populations [Fedorova 2013], consistent with the historical, archaeological and linguistic consensus, as well as mitochondrial lineage data [Zlojutro 2009]. While the precise timing of the waves of northward migration remains uncertain, it likely occurred no earlier than the 11th century and no later than the 15th century. The genetic homogeneity of the modern population has also been described as a consequence of a very strong bottleneck event associated with that northward migration [Pakendorf 2002, Pakendorf 2006]. Previous studies of archaeological material [Ricaud et al. 2006, Crubézy et al. 2010] have also shown that genetic diversity in the archaeological population was low and that there were few differences in the nature of Y-chromosome lineages between ancient and modern individuals.

The present study explores the demographic events that took place after the establishment of these Turkic populations in what is now called Central Yakutia.”

As it turns out from the Material and Methods, “The present study includes 23 unpublished ancient samples and 77 unpublished modern samples, collected during a dedicated expedition. This information remains hidden in the Abstract-Introduction-Result-Discussion-Conclusion part of the text. I missed from the Material and Methods the authenticity criteria of the ancient DNA results. How many times were these results reproduced? Have mtDNA analyses made the cleanness of the samples and validity of the results a certainty?”

The number of new samples is now mentioned in the Abstract and Introduction sections.

Along with repeated STR amplifications (at least two successful amplifications from two separate extractions), the mtDNA of five samples has been entirely sequenced. The quality of the sequences is very high, with average coverage depth between 1109x and 1461x and over 98% coverage uniformity. The exceptionally well-preserved nature of DNA from permafrost burials has allowed for successful Y-STR analysis in all but very few samples. These details are now presented in the Materials and Methods section, sub-section “DNA typing and sequencing” and the mtDNA results in a new table: “Supplementary Data 3”.

The paper neither compare the Y-STR data with other populations of Siberia nor mention Y haplogroups or Y-SNP based subgroup definitions. The study could certainly benefit from a deep Y –SNP test in the N1a (M46) branch.

The revised manuscript includes Y-haplogroups (that are presented in “Supplementary Data 1 - 340 Y-STR haplotypes.xlsx”) and a comparison with an in-house database of more than 200,000 Y-STR haplotypes (see “Results - Haplogroups and comparable lineages in other populations”). As previous studies have shown, we confirm the overwhelming dominance of haplogroup N1a, especially the N1a1-M46 branch. Below is the text added to the manuscript:

“Haplogroups and comparable lineages in other populations

Among the 21 different haplotypes identified in the ancient Yakut population, 17 belonged to the N1a1-M46 haplogroup (92% of individuals), 1 belonged to the N1a2-CTS6380 haplogroup (2 individuals), 1 to the C2-M217 haplogroup (2 individuals) and 1 to the C2b1a1b1-F3985 haplogroup (1 individual). Finally, one individual could not be affiliated reliably.

Searching for matching data in an in-house database containing more than 200,000 Y-STR haplotypes revealed that all 17 different N1a1-M46 haplotypes can be found (over 85% match, including 7 exact – 100% – matches) in the modern Yakut population. Among those, two are also found (including exact matches) in the modern Buryat population of southern Siberia. One of these two Buryat haplotypes belonged to an undated individual but the second one belonged to an individual buried before the 17th century in Central Yakutia. Results in ancient data (matches over 70%) were composed of diverse Mongol, Turkic and Yakut individuals.

The N1a2-CTS6380 haplotype, shared by two individuals anterior to the 17th century from Central Yakutia and the Vilyuy (western) region, was found (including 100% matches) in modern Khakassians. Results in ancient data were also composed of Mongol, Turkic and Yakut individuals.

The haplotype belonging to haplogroup C2-M217 was carried by two undated individuals buried close together in Central Yakutia. It matches (up to 100%) haplotypes found in the modern Buryat population and more than 100 modern Mongols. No ancient matches were found.

The individual belonging to haplogroup C2b1a1b1-F3985 was buried in Verkhoyansk before the 17th century. His haplotype was not found in any modern population but six matches over 70% were identified in ancient individuals from the European Steppe, possibly indicating an Altaic origin for this extinct lineage.

The one haplotype that could not be affiliated reliably (“Musée ethno”) was also undated and therefore not included in further analyses (no undated samples were). A matching haplotype (71%) was however found in an Early Okunevo individual. Since this analysis was performed on an individual presented in a museum with limited identification, these results could suggest a much earlier date for the skeleton (likely the Bronze Age).”

Line 166: Has the Yakut toyon Mazary Bozekov ever been analysed for Y –STR? That would be a good proof for the lines described in the first part of the Discussion („The expansion of the Ht1 is the expansion of the Kangalszy”).

Unfortunately, the grave of Bozekov was excavated in the 1930s and no element (bone, tooth or artefact) was collected at the time. It is supposed, from notes taken by researchers at the time, that the outline of the burial could still be found near the At Daban hills, where other individuals mentioned in this study (in particular At Daban 6) have been found, but it has not yet been recovered, although our work in the region continues. This has been made explicit in the text:

“While no skeletal elements were collected from the grave of Bozekov and therefore no genetic analyses performed, the notes taken by researchers at the time seem to indicate that the outline of the burial could still be found somewhere on Istiakh.”

Lines 196-198: What was the mitochondrial haplotype of the mother and his putative son? In supplement I only found the followings: “The individuals buried at Sytygane Syhé 1 and At Daban 6 share a mitochondrial HV-1 haplotype and belong 99 to haplogroup D5a2a.” Or is this mitochondrial analysis published already? If not, please describe the analyses in the Methods session and make available the sequence in NCBI/other database or the haplotype in a table. How frequent this haplotype among Yakuts?

The mitochondrial HV-1 haplotype for Sytygane Syhé 1 was published in Ricaut et al. 2006 whereas At Daban 6 was not analysed before the present study. We have added the relevant text to the materials and methods section and provide the HV-1 haplotype in Supplementary Data 2. Haplogroup D5a2a is in fact the dominant haplogroup in the ancient Yakut population, with 21.5% of carriers. The specific haplotype shared by At Daban 6 and Sytygane Syhé 1 is also quite common, with 14.6% of ancient Yakuts carrying this lineage.

The male buried at Sytygane Syhé 1 could be shown in more detail in the supplement. Has he had a rich grave? What kind of gravegoods has he had? It could be presented in the Supplement.

We did not present Sytygane Syhé 1 in the Supplement or the main text, because his grave was unfortunately poorly preserved and not photographed when it was excavated in 2002. It contained a complete skeleton that had been mostly moved to one corner (likely by water, rather than human or animal action) but no clothes had been preserved apart from a few beads (white and blue, typical of the early 18th century). The coffin had been reinforced with iron angles and the body had been wrapped in birch bark above and beneath. No other identifiable item was recovered.

Regrettably, while genetic analyses for Sytygane Syhé 1 were successful, the study of grave goods was limited beyond an estimation of the period of the burial (beads and iron angles being the main indicators).

Line 252: Reference to Tikhonov et al. 2019 is missing from the list of references. This study shows quite a few thematic overlap with the current paper under consideration, here own merits should be underlined through the distinction of the two studies.

We now discuss the Tikhonov study in more detail in the “Discussion - Mythological explanations and their limitations” section. While there was indeed overlap between their study and ours, the Tikhonov study was very interesting because it proposed arguments for the naming and biogeographical origin of the three main Y-lineages.

The forgotten reference was added to the list.

Why are some of the ancient Yakut graves presented in the supplement but most of them not mentioned? The supplementary table of 340 Y haplotypes include the ancient samples, but I don't see grave numbers or inventory numbers, any specification of these graves. Is Ebuguey 2 or Sola 2 unambiguously identify these specimens?

All names unambiguously identify one specimen. This has been made clear in the legends of the figures.

The graves presented in the supplementary files were meant to illustrate the evolution of archaeological material throughout the period and the diversity of grave goods. This has now been made clear in the text (Materials and Methods and first paragraph of Supplementary Material 1: archaeological phases).

Minor comments, ordered by occurrence in the main text:

Line 23: „... some of which had already been subjected to genetic analyses” Please specify, how many?

The number was 51 and it is now mentioned.

Line 49: „This work presents the rare case of ancient DNA studies leading to the identification of an individual and their place in the recent history of cultural evolution...” Their place of an individual? It is confusing.

This sentence has been amended. It now reads “her place in recent history” since the individual in question was At Daban 6, a woman.

Lines 59-64: I missed haplotype diversity for ancient individuals. How was the haplotype diversity calculated? Is it the Nei method?

We used the Nei method (1987). This reference has been added to the relevant section. We did not include haplotype diversity computed in this way for the ancient samples, since we lack exhaustive data concerning kinship and the ancient population spans five centuries. The figures for ancient data, that are not in the text, were 0.87 (diversity) and 12.16% (proportion of individuals carrying unique haplotypes). For the ancient samples, we instead focused on the demographic model in Table 2, that presents another method comparing numbers of different haplotypes in each period.

Lines 78-80: Please rephrase the sentence: „Other (minority) lineages...” I would use minor lineage instead of minority lineage through the text.

This has been corrected throughout the text, as suggested.

Figure 1a-c: highlighted site names are difficult to read, it would be good to read in the figure legends the total number of samples. Figure 1a-c: numbers after site names mean the number of samples? This should be explained in the legend.

The legends of these figures have been completed and clarified. Hopefully, the legibility of site names will be discussed with the editorial team during formatting. It is possible to use a larger font.

Lines 111-112: “while the Ht3 112 (western) line is dominant in the Vilyuy region (15/92 men, 16%).” It should be rephrased as: Ht3 112 (western) line is the most frequent in the Vilyuy region In Yakutia.

This has been corrected as suggested.

Lines 162: In table 2 the haplotype diversity numbers are very different from those written in Line 63. What is the methodical difference?

The term “diversity” created confusion in Table 2, since it is not the measure of diversity presented in the Materials and Methods section. The term has been replaced with “number of different haplotypes”.

Line 186: It would be good to have a larger figure (drawing) about the „The hagiographic ring” at least in supplement.

We have added a closeup and an impression to the supplementary material (Supplementary Material 4). We hope it will provide a better illustration of both signet rings. In a closeup, the Judgment of Solomon is unmistakable (Figure S8).

Lines 208: Cossacks and the Kangalaszy. It would be good if these two tribe (?) names were introduced earlier in the text, potentially in the Introduction, or detailed in the supplementary material 1 and cited somewhere earlier. In line 208 it is written: „..., then the most powerful Yakut tribes.” Were Cossack and Kangalaszy those? If yes, please rephrase this sentence (“who were the most powerful..”). If not, please add a sentence about Cossack and Kangalaszy tribes.

This passage has now been made clearer in the introduction and at line 208. The term “Cossacks” was used to refer to all Russians in Siberia in the 17th and 18th centuries, especially those carrying military service for the Russian state. Therefore, this sentence in the introduction has been completed:

“In the early 17th century, Cossacks serving the Russian Empire reached Siberia [...]”

And the sentence at line 208 has been amended as follows:

“[...] the period was marked by conflict between the Russian Cossacks, serving the Tsar, and the Kangalaszy, then the most powerful of Yakut tribes.”

The Kangalaszy tribe is also now mentioned in the Introduction.

Line 224: “In effect, male haplotypes underwent a bottleneck event, caused by the conflict between different lineage groups [Zeng et al. 2018]” The absolute dominance of the N-Tat Y haplogroups also shows a bottleneck among the Yakuts, a biased reproductive success which is described in previous papers that could be cited more explicitly here.

Yakut ethnogenesis was indeed marked by a significant bottleneck event that has been described in previous works. We now mention it more clearly in this section, as well as the beginning of the Conclusion.

Line 228: „presence of a lineage in Central Yakutia is anecdotal” Do you really mean that it was based on personal accounts rather than facts or research? I would write here simply unreliable instead of anecdotal. Or it simply hasn’t been detected yet among the 33 studied ancient individuals from this period (state of research).

Our choice of words was poor, and we have rewritten the sentence:

“(i) this line took hold in the Vilyuy before the Russian period and was in fact not widespread in Central Yakutia,”

This was meant to indicate that the presence of one individual belonging to the Ht3 line in Central Yakutia before the 18th century does not imply its establishment there.

Reviewers' comments:

Reviewer #1 (Remarks to the Author):

The authors examined ancient and modern Y-chromosomal lineages of Yakut with the goal of understanding demographic history. The manuscript presents genetic data first, then utilizes the information to expand specific archaeological questions and cultural-historical context of social structure. There are broader questions and some structural issues, as well as specific content that appears missing.

1. While the historical context is intriguing, how is this information relevant and meaningful to a general scientific audience beyond those who already have an interest in Siberian history and population genetics? Why do the results matter and why should people care? This question is not adequately highlighted or emphasized in an effective manner to appeal to a broader audience.
2. The manuscript refers to the previous studies regarding the samples, but it should be at least briefly highlighted where these ancient samples come from without the reader having to look up those studies. From which site did they come from, what are the dates, what is the context?
3. For the unpublished samples (23 ancient), how were they obtained? What sites do they come from, what are the dates, etc? There is no explanation in either results or methods- if these samples are unpublished, a full report should be included from excavation, sexing, DNA extraction, etc.
4. The archaeological investigation comes as an abrupt shift when the paper gives the impression as largely a population genetics investigation. This may be an organizational issue, the manuscript either needs a much better setting up for the questions in the introduction or change the order of information presented to clearly distinguish the various methods employed. Specifically, the section regarding At-Daban 6 grave does not seem to flow with the preceding information.
5. The discussion of preferential burial is also unexpected and seemed random when almost any information regarding excavation and archaeological investigation was absent in the preceding pages.
6. When the authors speak about Yakut population genetics, why not speak about broader Siberian Y-diversity and previous studies to better understand the context? How does the Yakut compare to other Siberian populations, either past or present? How does it compare to the mtDNA diversity, ancient and modern? It isn't clear how this research adds to current understandings of Siberian population genetics, and whether the "very low" diversity is unique. Low Y-diversity has been observed in other populations, what does it mean by "very low"? (Line 74) Can this also be an artifact of sampling?
7. Discussion regarding the mythological narratives are certainly central to the study's goals - perhaps this needs to be explained in a little more detail in the Introduction to properly set-up and highlight why the study is important and the specific questions it can address. The information presented in the introduction is very vague and provides no specific timeline or previous study of two dominant lineages (which authors elude to in the Discussion, Lines 243-260). Parts of the discussion may be better served in the Introduction, which then later the authors could better highlight how their study adds to this view.
8. A discussion on the limitations of the study should be included, namely whether the 74 men can be reflective of the ancient population.

Reviewer #3 (Remarks to the Author):

The authors addressed all my comments and questions. I believe that the paper has been improved significantly, and now I suggest its acceptance and publication.

Reviewer #1 (Remarks to the Author):

The authors examined ancient and modern Y-chromosomal lineages of Yakut with the goal of understanding demographic history. The manuscript presents genetic data first, then utilizes the information to expand specific archaeological questions and cultural-historical context of social structure. There are broader questions and some structural issues, as well as specific content that appears missing.

The changes introduced in the text and described below emphasize the three main points that are of interest to a broader audience, namely (1) the unprecedented demonstration of burial preference/bias based on lineage in an ancient population, (2) the genetic dynamics of colonisation that led to the demographic and cultural expansion of the conquered rather than the conquerors and (3) the unusually successful confrontation of mythological/historical data and genetic data, which allowed us to gain insight into recent cultural constructs.

The introduction was especially revised, to better provide a framework for following the different analyses that were performed.

1. While the historical context is intriguing, how is this information relevant and meaningful to a general scientific audience beyond those who already have an interest in Siberian history and population genetics? Why do the results matter and why should people care? This question is not adequately highlighted or emphasized in an effective manner to appeal to a broader audience.

We have added text underlining the most significant aspects of the study and we have especially rewritten the Abstract to include them:

“Abstract

Seventeen years of archaeological and anthropological expeditions in North-Eastern Siberia (in the Sakha Republic, Yakutia) have permitted the genetic analysis of 150 ancient (15th-19th century) and 510 modern individuals. Almost all males were successfully analysed (Y-STR) and this allowed us to identify paternal lineages and their geographical expansion through time. This genetic data was confronted with mythological, historical and material evidence to establish the sequence of events that built the modern Yakut genetic diversity.

We show that the ancient Yakuts recovered from this large collection of graves are not representative of an ancient population. Uncommonly, we were also able to demonstrate that the funerary preference observed here involved three specific male lineages, especially in the 18th century. Moreover, this dominance was likely caused by the Russian conquest of Siberia which allowed some male clans to rise to new levels of power. Finally, we give indications that some mythical and historical figures might have been the actors of those genetic changes.

These results help us reconsider the genetic dynamics of colonization in some regions, question the distinction between fact and myth in national histories and provide a rare insight into a funerary ensemble by revealing the biased process of its composition.”

2. The manuscript refers to the previous studies regarding the samples, but it should be at least briefly highlighted where these ancient samples come from without the reader having to look up those studies. From which site did they come from, what are the dates, what is the context?

A new subsection has been added to the Materials and Methods describing the general characteristics of the field and archaeological samples and points to the details presented in

Supplementary Material 1. The revised introduction also mentions the archaeological context and the dating.

3. For the unpublished samples (23 ancient), how were they obtained? What sites do they come from, what are the dates, etc? There is no explanation in either results or methods- if these samples are unpublished, a full report should be included from excavation, sexing, DNA extraction, etc.

The new subsection of the Materials and Methods mentioned in the answer to the previous question now presents the characteristics of previously unpublished archaeological samples.

4. The archaeological investigation comes as an abrupt shift when the paper gives the impression as largely a population genetics investigation. This may be an organizational issue, the manuscript either needs a much better setting up for the questions in the introduction or change the order of information presented to clearly distinguish the various methods employed. Specifically, the section regarding At-Daban 6 grave does not seem to flow with the preceding information.

The revised Introduction includes two new paragraphs that should expose more clearly the rationale behind the sequence in which we presented the results:

“The present study explores the demographic events that took place after the establishment of these Turkic populations in what is now called Central Yakutia. It has been evident from the first studies that this large collection of Yakut graves was not representative of the ancient population. It is not a set of cemeteries, where the deceased might be routinely buried, since such practices were almost inexistent in medieval Yakutia, where burial itself is a difficult and time-consuming task because of the frozen ground. It is rather a series of isolated graves, with some archaeological sites discovered tens of kilometres away from any other grave. The homogeneity of grave goods does not extend beyond a few very characteristically Yakut elements which evolved recognisably throughout the period and adapted to different biomes. Most graves present one or more original elements that make finding a common classifier a difficult task. Since mitochondrial diversity seems to be maintained throughout the period in all regions [6], we endeavoured to determine whether Y-chromosome lineage diversity could be linked to the selection of individuals for burial, rather than perishable disposal, as was usual. We therefore identified lineages and placed them in a chronological and geographical context to better understand how the paternal lineages represented in the archaeological record conferred burial privileges to the individuals who carried them.

Having identified favoured lineages, we focused on the graves that were associated with them to determine whether they contained other signs of high social status or reasons for preferential burial. While some burials could be explained by epidemiological or religious factors (shamanic or Christian), the presence of Russian-made artifacts led us to focus on a few graves, especially the grave of a woman buried on the known territory of a powerful medieval clan. This gave us the opportunity to explore historical and archival data associated with the Russian conquest of Siberia that seemed to directly link the rise to power of a local tribe (the Kangalasy) to that specific grave, while genetic data indicated a link between the occupant of the grave and the dominant Y-chromosome lineage.”

5. The discussion of preferential burial is also unexpected and seemed random when almost any information regarding excavation and archaeological investigation was absent in the preceding pages.

The revised Introduction should clarify the reasons why preferential burial is an integral part of the discussion.

6. When the authors speak about Yakut population genetics, why not speak about broader Siberian Y-diversity and previous studies to better understand the context? How does the Yakut compare to other Siberian populations, either past or present? How does it compare to the mtDNA diversity, ancient and modern? It isn't clear how this research adds to current understandings of Siberian population genetics, and whether the "very low" diversity is unique. Low Y-diversity has been observed in other populations, what does it mean by "very low"? (Line 74) Can this also be an artifact of sampling?

The revised text now clarifies that the ancient sample set should not be expected to represent the ancient population, given evident biases in its constitution. We do not discuss Y-chromosome diversity in other Siberian populations because recent data (using at least the same number of Y-STR loci) is scarce for these regions. The literature however makes it clear that the Yakuts constitute one of the least diverse populations of Siberia when it comes to lineages. Concerning modern data, we do not believe that the low diversity could be an artefact of sampling given the geographical range of collection campaigns in the Sakha Republic.

7. Discussion regarding the mythological narratives are certainly central to the study's goals - perhaps this needs to be explained in a little more detail in the Introduction to properly set-up and highlight why the study is important and the specific questions it can address. The information presented in the introduction is very vague and provides no specific timeline or previous study of two dominant lineages (which authors elude to in the Discussion, Lines 243-260). Parts of the discussion may be better served in the Introduction, which then later the authors could better highlight how their study adds to this view.

The Abstract has been completely revised and two paragraphs have been added to the Introduction to set-up the three important elements we believe this study sheds light on. We hope the changes now expose more clearly the study's goals: (1) provide a specific explanation for the biased constitution of the funerary ensemble (burial preference for the elite), (2) explain the dominance of an otherwise defeated ethnic group and (3) identify what links exist between mythical histories and actual demographic or historical events.

8. A discussion on the limitations of the study should be included, namely whether the 74 men can be reflective of the ancient population.

It is likely that the 74 men are not reflective of the ancient population. The new introduction to the discussion of funerary preference hopefully clarifies that we did not intend to present the samples in this way.

Reviewer #3 (Remarks to the Author):

The authors addressed all my comments and questions. I believe that the paper has been improved significantly, and now I suggest its acceptance and publication.

We would like to thank the reviewer for their continued work.